# Incidence and Risk Factors of Hepatocellular Carcinoma in Patients with Chronic Hepatitis C Treated with Direct-Acting Antivirals

**DOI:** 10.3390/v15010221

**Published:** 2023-01-13

**Authors:** Cassia Leal, Jorge Strogoff-de-Matos, Carmem Theodoro, Rosangela Teixeira, Renata Perez, Thais Guaraná, Paulo de Tarso Pinto, Tatiana Guimarães, Solange Artimos

**Affiliations:** 1Gastroenterology and Hepatology Unit, Internal Medicine Department, Hospital Federal dos Servidores do Estado, Rio de Janeiro 20221-161, Brazil; 2Gastroenterology and Hepatology Unit, Antônio Pedro Universitary Hospital, Fluminense Federal University, Rio de Janeiro 24033-900, Brazil; 3Departamento de Medicina Clínica, Faculdade de Medicina, Universidade Federal Fluminense, Rio de Janeiro 24033-900, Brazil; 4Departamento de Clínica Médica, Faculdade de Medicina, Instituto Alfa de Gastroenterologia Hospital das Clínicas, Universidade Federal de Minas Gerais, Belo Horizonte 30130-100, Brazil; 5Hospital Felício Rocho, Belo Horizonte 30110-934, Brazil; 6Hepatology Division, D’Or Institute for Research and Education (IDOR), Rio de Janeiro 22281-100, Brazil; 7Hepatology Division, Federal University of Rio de Janeiro, Rio de Janeiro 21941913, Brazil; 8Departamento Materno Infantil, Faculdade de Medicina, Universidade Federal Fluminense, Rio de Janeiro 24033-900, Brazil

**Keywords:** hepatocellular carcinoma, incidence, hepatitis C treatment, direct-acting antivirals

## Abstract

Background: Conflicting data regarding the incidence of hepatocellular carcinoma (HCC) after cure of HCV infection with direct-acting antivirals (DAAs) remains. We investigated the incidence and risk factors to HCC after treatment with DAAs followed up for five years. Methods: A total of 1075 HCV patients ≥ 18 years were treated with DAAs from 2015 to 2019 and followed until 2022. Ultrasonography was performed before DAAs and each 6 months thereafter. Results: Of the total, 51/1075 (4.7%) developed HCC in the median of 40 (IQR 25–58) months: 26/51 (51%) male, median age 60 (IQR 54–66) years, alpha-fetoprotein (AFP) 12.2 (IQR 6.1–18.8) ng/mL, 47/51 (92.1%) cirrhotic 78.7%, 8/51 (15.7%) without sustained virological response (SVR). Seventeen percent had non-characterized nodules before DAAs. Cumulative HCC incidence was 5.9% in 5 years. Overall incidence was 1.46/100 patient-years (PY) (95% CI = 1.09–1.91), being 2.31/100 PY (95% CI = 1.70–3.06), 0.45/100 PY (95% CI = 0.09–1.32) and 0.20/100 PY (95% CI 0.01–1.01) in METAVIR F4, F3 and F2, respectively, and the main risks to HCC were non-characterized nodule, cirrhosis, high AFP values and non-SVR. Conclusion: HCV cure reduced risk for HCC, but it still occurred particularly in cirrhotic patients. Some risk factors can be identified to predict early HCC diagnosis.

## 1. Introduction

Liver cirrhosis caused by hepatitis C virus (HCV) is currently the leading cause of hepatocellular carcinoma (HCC) worldwide [1], with an annual rate of 1% to 4% [2]. The cure of HCV infection with new direct-acting antivirals (DAAs) surpasses 90% [3], with remarkable benefits including improvement of liver function, decrease of portal hypertension [4,5], lower rates of hospitalization [6] and of liver transplantation [7].

Despite unequivocal evidence of the benefits of the HCV cure, concerns have been raised that the treatment of HCV with DAAs might induce the growth of pre-existing tumor cell clones or the emergence of new tumor cells by triggering abrupt changes in the relationship between the inflammatory state and immune stimulation [8]. Therefore, there is debate that HCV cure could be associated with the recurrence and development of HCC [9,10,11,12,13,14,15]. 

The risk of HCC is lower in patients with chronic HCV infection without established cirrhosis [16,17], though surveillance of HCC in patients with advanced liver fibrosis (METAVIR score ≥ F3) [18] is still recommended [19]. However, current evidence is scarce and based on retrospective studies with small cohorts [17]. 

In view of the controversies about potential association between HCC and HCV treatment with DAAs in patients with advanced fibrosis, we investigated the incidence of HCC in a large cohort of patients with chronic hepatitis C treated with DAAs and followed up for a long period.

## 2. Materials and Methods

This is a prospective cohort study conducted at two tertiary hospitals in Brazil that apply equal protocols of HCV treatment. Eligibility criteria were patients ≥ 18 years of age with chronic hepatitis C, regardless of liver fibrosis stage, treated with DAAs from November 2015 to July 2019 (inclusion period). Patients were followed up prospectively after the end of treatment until February 2022.

The inclusion criteria were (i) treatment with DAAs for at least 4 weeks, irrespective of previous treatment with any scheme of interferon (IFN) based therapy; (ii) assessment by abdominal imaging (ultrasound (US), computed tomography (CT) or magnetic resonance imaging (MRI)) up to six months before starting DAAs. Exclusion criteria were (i) diagnosis of HCC before starting DAAs; (ii) liver transplant prior to treatment with DAAs; (iii) treatment of HCV with combined IFN and DAAs. All eligible patients who agreed voluntarily to participate were included.

The treatment of hepatitis C with DAAs followed the Clinical Protocol and Therapeutic Guidelines for Hepatitis C and Coinfections of the Brazilian Ministry of Health [20]. If patients received sequential DAA treatment after inclusion, the date of the first DAA treatment was defined as “time zero”. Data on subsequent DAA treatments and the responses obtained were not included in the study. Sustained virological response (SVR) was defined as undetectable HCV RNA by quantitative polymerase chain reaction 12 weeks after the end of treatment.

Abdominal US was performed up to six months before the start of DAAs and at intervals of six months as a protocol for participants with advanced fibrosis (F3 or F4) (or according to clinical decision for others) during and after DAA treatment. If abdominal US detected any focal liver lesion, abdominal CT or MRI was performed for more accurate diagnosis.

The liver nodules detected before DAA treatment were classified as “characterized nodule” (definitively benign nodule), or “non-characterized nodule” (nodules ≤ 10 mm or > 10 mm in which the diagnosis of HCC was carefully ruled out by CT, MRI, or biopsy) [8,14] before starting DAAs. 

Liver fibrosis was staged before treatment by liver biopsy (when available) and/or transient elastography (TE) (FibroScan^®^ Echosens, Paris, France), which was repeated once 6 to 12 months after the end of treatment. Diagnostic criteria of cirrhosis were liver biopsy classified as METAVIR F4, liver stiffness measurement (LSM) > 12.5 kPa, presence of portal hypertension (esophageal/gastric varices), previous liver decompensation, or at least two abdominal US criteria such as liver surface irregularity, enlarged spleen or portal vein diameter > 12 mm. 

Clinically significant portal hypertension was defined as TE (LSM > 25 kPa, between 20–25 kPa and platelet count < 150 × 10^9^/L or 15–20 kPa and platelet count < 110 × 10^9^/L) [21], previous decompensation or presence of esophageal/gastric varices. Patients without clinical/radiologic signs of cirrhosis before HCV treatment were defined based on LSM value (METAVIR score): ≤7 kPa (F0–F1); 7.1–9.5 kPa (F2); 9.6–12.5 kPa (F3) and >12.5 kPa (F4) [22].

### 2.1. Demographic, Clinical and Laboratory Data

All data were collected from medical records. Baseline and demographic data included age, gender, body mass index, Child–Pugh score, comorbidities (diabetes mellitus, previous history of cancer, alcohol use (daily alcohol consumption ≥30 g for women and ≥40 g for men) [23], coinfection with hepatitis B virus (HBV) and/or human immunodeficiency virus (HIV)).

Laboratory and virological parameters included platelet count, aminotransferases, albumin, bilirubin, international normalized ratio (INR), alpha-fetoprotein (AFP), AST to platelet ratio index (APRI), fibrosis-4 (FIB-4) index, albumin–bilirubin (ALBI) score, HCV genotype and viral load. Abdominal US, TE, triphasic abdominal CT or MRI data were also collected. Data also included clinical records of previous antiviral treatment, type of DAA therapy and response to DAA treatment. 

### 2.2. Outcome Assessment

Patients were prospectively followed, and data were collected until the development of HCC, death or loss to follow-up. For patients who did not reach those outcomes, data were censored at 60 months of follow-up or on 28 February 2022. For patients in whom a liver nodule was detected by abdominal US during the follow-up, the subsequent procedures followed current guidelines. If a nodule was < 10 mm, abdominal US was performed every 3 months; if a nodule was ≥ 10 mm, abdominal CT or MRI with dynamic analysis of the liver was performed.

The diagnosis of HCC was established based on the criteria of the European Association for the Study of the Liver (EASL) [19]. In the case of an incidental nodule with initial exclusion of HCC (by CT/MRI or biopsy), the follow-up interval was reduced to three months for a period of 24 months and, thereafter, back to six months if no change in the size or number of nodules were observed. Patients diagnosed with hepatic tumor other than HCC during the follow-up were referred to the oncology service.

The following parameters were evaluated at the diagnosis of HCC according to the Barcelona Clinic Liver Cancer (BCLC) staging system [24]: liver profile, Child–Pugh classification, tumor burden, tumor-related symptoms (Eastern Cooperative Oncology Group Performance Status—ECOG-PS).

### 2.3. Temporal Definitions for Clinical Assessment

Temporal definitions were proposed for the study, such as the following: (i) interval between the last pre-DAA image and the start date of DAA (up to six months); (ii) interval between the start of antiviral treatment (“time zero”) and the last post-DAA image; (iii) interval between the start of DAA and diagnosis of HCC. 

### 2.4. Statistical Analysis

Variables with normal and non-normal distribution were expressed as mean ± standard deviation, and median and interquartile range (IQR), respectively. Quantitative variables were compared using appropriate tests. The incidence density was expressed as the number of events per one hundred patient-years and 95% confidence interval. Survival curves used the Kaplan–Meier method and the log rank test. Cox proportional hazard models analyzed the association between the development of HCC after DAA treatment. Variables with greater biological plausibility of association with the development of HCC were included in the multivariate model. A *p* value < 0.05 was considered statistically significant. The sensitivity and specificity of TE, ALBI, FIB-4 and APRI scores as predictors of HCC development were evaluated through the calculation of the area under the receiver operating characteristics (ROC) curve. The point at which the sum of sensitivity and specificity was maximized was considered the optimal cutoff value according to Younden’s method [25]. The areas under ROC curves were compared using MedCalc18.6 (MedCalc Software, Ostend, Belgium). Statistical analysis was performed using the program SPSS 21.0 (IBM©, Chicago, IL, USA). 

## 3. Results

### 3.1. Baseline Characteristics of Participants

A cohort of 1211 patients with chronic hepatitis C were prospectively treated with DAAs from November 2015 to July 2019. Of these, 136 (11.2%) were excluded and 1075 met the inclusion/exclusion criteria and were included in the study (Figure 1). 

HCV: hepatitis C virus; HCC: hepatocellular carcinoma; DAA: direct antiviral agent; US: ultrasound; IFN: interferon; LT: liver transplantation, SVR: sustained virologic response.

Six hundred and forty (59.5%) patients were female. The median age was 60 years (IQR 53–66). Five hundred and ninety-eight (55.6%) patients were diagnosed with cirrhosis, being 85.6%, 13.9% and 0.5% classified as Child–Pugh A, B and C, respectively. 

The median AFP was 5.5 (IQR 3.3–10.2). Two hundred forty-five (22.8%) had AFP higher than 10 ng/mL. Three hundred twenty-eight (30.5%) had portal hypertension, and 98.7%, 0.8% and 0.2% patients had an ALBI score of 1, 2 and 3, respectively. 

The median interval between the last exam of image before the start of DAA was 1.3 (IQR 0.7–2.1) months. Sixty-eight (6.3%) patients had liver nodules detected before treatment, with 17 (1.6%) classified as non-characterized nodules. Baseline demographic, clinical and laboratory data are presented in Table 1.

### 3.2. HCV Treatment

One thousand seventy-five patients were treated with DAAs: 678 (63.1%) with sofosbuvir and daclatasvir, and 264 (24.5%) with simeprevir with or without ribavirin. The duration of treatment was 12 weeks in 925 (86%) patients. The rate of SVR was 97.4% (*n* = 1047). Three hundred and ninety-six patients (36.8%) were non-responders to previous HCV treatment.

### 3.3. Incidence of Hepatocellular Carcinoma

Fifty-one (4.7%) patients developed HCC during the median follow-up after treatment of 40.3 (IQR 25.5–57.7) months. The annual incidence during the period of follow-up was 1.7%, 1.4%, 1.2%, 1.3% and 0.9% after 1, 2, 3, 4 and 5 years, respectively. The cumulative incidence of HCC was 5.9% after 5 years (Figure 2).

The median of the interval between the start of DAA treatment and the diagnosis of HCC was 17.9 (IQR 5.8–33.4) months.

Among the 51 patients who developed HCC, 26 (51%) were male with a median of age 60 (IQR 54–66) years and had serum AFP level of 12.2 (IQR 6.1–18.8) ng/mL at the start of DAA treatment. Forty-seven (92.1%) had the diagnosis of cirrhosis, being 78.7% and 21.3% classified as Child–Pugh A and B, respectively. Regarding HCV treatment with DAAs, eight (15.7%) patients who developed HCC did not achieve SVR.

At diagnosis of HCC, ECOG-PS was 0, 1 and 3 in 44 (86.3%), 6 (11.8%) and in 1 (1.9%) patient, respectively. In addition, 49 (96.1%) and 2 (3.9%) patients were graded as ALBI 1 and ALBI 2, respectively. Thirteen (25.5%) out of 51 participants who developed HCC during follow up had liver nodules diagnosed before DAA treatment, nine (17.6%) having non-characterized nodules.

In our study population, 17 (1.6%) patients had non-characterized nodules prior to DAA therapy, with low serum AFP levels. Such nodules were surveyed by repeated imaging (CT or MRI) in all 17 patients, at 3-month intervals, before starting HCV treatment, and did not present typical HCC findings. The majority of non-characterized nodules were dysplastic nodules, i.e., nodules which were not cyst, hemangioma, focal nodular hyperplasia, adenoma or HCC [26]. If there was a change in the radiologic pattern or an increase in AFP levels, the patient underwent a biopsy, which was necessary in two (11.8%) patients. Overall, nine (52.9%) patients with non-characterized nodules developed HCC during follow-up, and in six (35.3%) patients, HCC occurred within the non-characterized nodule.

According to BCLC [24], liver tumors were classified as follows: 0 (8/51, 15.7%), A (26/51, 51%), B (11/51, 21.6%), C (5/51, 9.8%) and D (1/51, 1.9%).

The global incidence of HCC was 1.46/100 patient-years (95% CI 1.09–1.91). According to the stage of liver fibrosis (METAVIR F0–4), the incidence rates were: (i) 2.31/100 patient-years (95% CI 1.70–3.06) in F4; (ii) 0.45/100 patient-years (95% CI 0.09–1.32) in F3; (iii) 0.20/100 patient-years (95% CI 0.01–1.01) in F2. No case of HCC was diagnosed among F1/F0 patients.

### 3.4. Risk Factors for the Development of HCC

Table 2 shows the factors potentially associated with increased risk of development of HCC after the start of DAA treatment. By univariate analysis, male gender, age, alcohol use, high LSM values, presence of non-characterized liver nodule, presence of ascites, time since pretreatment imaging, serum albumin < 3.5 g/dL, platelet count < 100,000 mm^3^, AFP > 10 ng/mL, high INR, elevated bilirubin and alkaline phosphatase, genotype 1, and lack of response to DAA treatment were significantly associated with development of HCC after DAA treatment.

The multivariate analysis included variables with *p* < 0.20 [27] after removing overlapping or interacting variables associated with the severity of cirrhosis. In the adjusted model, age, presence of a non-characterized nodule, high LSM values, high AFP values and non-SVR remained as risk factors for the development of HCC (Table 3).

Analysis of the 5-year cumulative incidence showed a significantly higher incidence of HCC among patients with cirrhosis compared to those without cirrhosis (9.7% vs. 1.6%, *p* < 0.0001), Figure 3a. In addition, patients who failed to achieve SVR also had higher incidence of HCC as compared to those who did (43% and 5.8%, *p* < 0.0001) (Figure 3b).

Of note, we found a remarkably higher cumulative incidence of HCC among patients with non-characterized hepatic nodules in the pre-treatment period. Hence, the cumulative incidence of HCC was 29.4% in patients with non-characterized nodules and 4.0% in patients with characterized nodules in the first 12 months after the start of DAAs. After 5 years, these rates were 60.3% and 6.3%, respectively (*p* < 0.0001) (Figure 3c).

The ROC curve analysis was performed to identify predictors of the development of HCC using TE, ALBI, FIB-4 and APRI scores before the start of DAA treatment (Figure 4). The largest area under the curve (AUC) was observed for TE (AUC: 0.79; 95% CI 0.73–0.85) with sensitivity of 86% and specificity of 61% for the cutoff of 14.3 kPa, followed by FIB-4 (AUC: 0.72; 95% CI 0.66–0.78) with sensitivity of 76% and specificity of 63%.

After comparing the ROC curves using DeLong test, we concluded that TE has better predictability than ALBI, FIB-4 and APRI. The area under the ROC curve for TE was higher than the APRI (*p* = 0.001), FIB-4 (*p* = 0.017) and ALBI (*p* = 0.011) scores, without statistical significance between APRI, FIB-4 and ALBI.

With a cutoff of 20 kPa, 25 kPa and 30 kPa on the EHT ROC curve, we obtained a sensitivity of 67% with a specificity of 75%, 44.4% with 82.6% and 20.8% with 87.5%, respectively.

With the cutoff of our study of 14.3 kPa (sensibility of 86% and specificity of 61%) and considering the cutoff of Baveno VII of 20 kPa [21], the incidence of HCC in 5 years for LSM <15 or ≥15 kPa and <20 or ≥20 kPa were 2.1% or 9.7% and 2.8% or 11.4%, respectively (Figure 5).

### 3.5. Survival

Sixty-nine (6.4%) patients died during the study period, including 23 (33.3%) due to causes related to liver cirrhosis and 16 (23.2%) because of progression of HCC.

## 4. Discussion

The association between the new DAA treatment for HCV and the appearance of HCC is an unsolved issue that has been widely debated recently. We presented here results of our prospective investigation that included a cohort of 1075 patients with chronic hepatitis C treated with DAAs followed for a median time of 40.3 months after treatment. Fifty-one (4.7%) developed HCC, with 67% of tumors diagnosed in early stages (BCLC 0: 16% and BCLC A: 51%). The median interval between the start of DAA treatment and the diagnosis of HCC was of 17.9 months.

The global incidence of HCC was 1.46/100 patient-years. As expected, the incidence rate was higher in cirrhotic patients (2.31/100 patient-years), with substantially lower incidence in patients with fibrosis stage METAVIR F3 (0.45/100 patient-years) and F2 (0.20/100 patient-years). Of note, we observed that there was development of HCC in patients with fibrosis METAVIR F2, confirmed by liver and tumor biopsy, although with a low incidence.

The highest annual incidence occurred in the first year after HCV treatment, with reducing incidence in subsequent years (annual incidence: 1.7%, 1.4%, 1.2%, 1.3% and 0.9% after 1, 2, 3, 4 and 5 years, respectively). Hence, our data suggest a yearly decline in the incidence rate of HCC after the HCV cure.

Some authors have suggested that the reason for the decay in the incidence of HCC with time is related to the regression of liver fibrosis, which is a slow process after HCV eradication [28]. However, there is great controversy in the literature regarding the reduction in the risk of HCC after treatment with DAAs, as it is currently unclear if the HCC risk truly declines over time after HCV eradication. Thus, it is currently recommended that cirrhotic patients should have indefinite surveillance after HCV cure.

Although retrospective, the current most convincing evidence regarding lower risk of HCC after HCV cure in patients with cirrhosis came from the ERCHIVES study that analyzed a cohort of 17,836 patients with cirrhosis and revealed a much lower incidence of HCC in patients with cirrhosis who achieved SVR following DAA (2.12/100 PY) or interferon treatment (2.28/100 PY) compared to untreated cirrhotic patients (4.53/100 PY) (*p* = 0.03) [29]. Therefore, treatment with DAAs did not represent a risk for the development of HCC in patients with cirrhosis, and SVR has been consolidated as the most important determinant of a lower HCC risk so far.

This fact brings up another critical issue to the debate. As the number of patients with HCV cure continues to increase, it is important to identify which patients could benefit from ongoing HCC surveillance, since it is recommended only for populations where the HCC incidence exceeds 1.5% per year [19,28].

In our prospective study, we demonstrated a global HCC incidence of 1.46/100 PY in patients following DAA treatment, being 2.31/100 PY in cirrhotic patients and 0.45/100 PY in patients with advanced fibrosis (F3). Therefore, our study shows that the incidence of HCC in patients with fibrosis METAVIR F3 is lower than in cirrhotic patients after treatment with DAAs. Similar findings were also reported in a recent systematic review and meta-analysis that identified forty-four studies (107,548 person-years of follow-up), where the incidence of HCC was 2.10/100 PY (95% CI, 1.9–2.4) among patients with cirrhosis and 0.50/100 PY (95% CI, 0.3–0.7) among patients with F3 fibrosis [28]. Comparable results were also reported in a recent small series [16,17].

Thus, for patients with advanced fibrosis (METAVIR F3), guidelines are inconsistent in their recommendations, as the lower incidence of HCC after HCV treatment, compared to patients with cirrhosis, is below the recommended threshold for cost-effective screening.

The current recommendations of the major international guidelines regarding HCC surveillance in F3 patients are still weak and with a low level of evidence. The American Association for the Study of Liver Disease (AASLD) guideline [30] proposes HCC surveillance be optional in this group of patients, since the incidence of HCC is lower than 1.5% per year. While there is proven benefit of HCC surveillance in F4 patients, this is uncertain for F3 patients. The European Association for the Study of the Liver (EASL) guideline [19] suggests that non-cirrhotic F3 patients may be considered for surveillance, but also does not advise surveillance due to low level of evidence (weak recommendation).

Furthermore, Farhang Zangneh et al. [31] recently evaluated the cost-effectiveness of HCC screening based on bi-annual abdominal ultrasound in patients with F3 fibrosis after SVR and found an incremental cost-effectiveness ratio of USD 188,157 per quality-adjusted life year, indicating that screening is not cost-effective in this group of patients, reflecting in an excessive burden for the health system.

Given the information presented above, the benefit of HCC surveillance after SVR in F3 patients continues to create uncertainty, therefore requiring more evidence.

Our study also evaluated the risk for HCC in patients with F3 in a large cohort followed up over a longer period, contributing to this current controversy. Combined with the results of studies published since 2018, our findings encourage clinicians to withdraw F3 patients from HCC screening programs safely if they do not have additional HCC risk factors. We believe that screening for HCC in F3 patients with SVR could probably be modified.

This issue is truly relevant in clinical practice, especially if we consider that the routine ultrasound screening for all patients who really need it implies an overload of exams in public services and demand for indefinite need of adherence for patients during their life. Thus, excluding F3 patients would increase the availability of resources for more effective screening of F4 patients.

With these considerations, one great challenge so far is to find methods to identify which patients are at higher risk of developing HCC after SVR.

APRI and FIB-4 scores have been used to assess the HCC risks. However, they were not developed specifically for prediction of HCC and have limited accuracy. Similarly, transient elastography (TE) was not designed to detect HCC. Since the pioneering study from Masuzaki and colleagues [32], controversy remains regarding the grade of the liver stiffness, using TE, for predicting HCC risk.

As there are no validated predictive risk models in this population so far, we included the TE, ALBI, APRI and FIB-4 scores in our study. The best performance to predict HCC was the TE, with cutoff points of 14.3 kPa and 20 kPa. In fact, the five years incidence of HCC increased with the higher values of liver stiffness measurement (LSM): 9.7% for TE ≥ 15 kPa and 11.4% for TE ≥ 20 kPa. Another prospective cohort (*n* = 99) reported the best cutoff LSM value of > 21.1 kPa (HR: 5.548; 95% CI: 1.244–24.766; *p* = 0.025), reiterating the magnitude of LSM as predictive of HCC in patients with cirrhosis due to HCV [32].

Put together, the results suggest that the higher the LSM, the greater the risk attributed. Hence, in the light of current knowledge, patients with LSM by TE > 20 kPa at baseline should be rigorously included on HCC surveillance programs and followed actively for adherence. LSM needs to be undertaken before HCV treatment, as the values usually decrease and might not apply for treated patients after successful HCV eradication [33].

Another challenge in cirrhotic patients in clinical practice is the detection of undetermined nodules in the pre-DAA period that turn into overt HCC during or early after treatment. In our cohort, we observed a remarkably higher cumulative incidence of HCC among patients with non-characterized hepatic nodules in the pre-treatment period. A similar finding was also reported by Mariño and colleagues [14], in their retrospective multicenter study of 1123 cirrhotic patients treated with DAAs, where the existence of indeterminate nodules before DAA treatment was associated with a three-time higher risk of HCC (HR = 2.83; 95% CI 1.55–5.16).

The nodules could be dysplastic or even the disturbed architecture of the cirrhotic liver. Thus, while most non-characterized nodules are expected to correspond to low or high-grade dysplastic foci or macro regenerative nodules in a cirrhotic liver [14], in some cases, as an infrequent circumstance, it may have been initial HCC or nodules displaying preexisting or microscopically undetectable tumors, as suggested by Romano and colleagues [34]. Data from a retrospective Italian study with 36 patients with dysplastic nodule classified by biopsy (21 low-grade, 15 high-grade, 17.4 ± 2.6 mm) showed that this may happen after the median of 36 (6–128) months of follow-up, being more often in high- versus low-grade dysplastic nodules (32.2% vs. 9.3% per year, *p* = 0.0039) [35].

A further point argued by others is that the development of HCC in DAA-treated patients could correspond to the emergence of already existing clones and even appear irrespective of treatment. The mechanisms involved in the transition from dysplasia to HCC or the growth of dormant clones are not well understood, but immune surveillance is surely involved [8,10].

The fact that in our result, a non-characterized nodule was associated with an almost 30% cumulative chance of development of HCC in the first 12 months after DAA treatment clearly suggests that these patients need a closer follow-up.

AFP has been useful in the diagnosis of HCC in clinical practice but lacks specificity as it is often elevated in cirrhotic patients without HCC, in the presence of non-HCC tumors (such as testicular germinal tumors, cholangiocarcinoma and gastric adenocarcinoma) and during liver regeneration following hepatic resection or recovery from massive hepatic necrosis [35]. Kumada and colleagues [36] reported that AFP ≥ 5.0 ng/mL is independently associated with the development of HCC within 10 years after SVR.

In our investigation, we found an association of high AFP levels before DAA as an independent predictor of the incidence of HCC. There are no other recent prospective studies based on DAA treatments that predict the risk of HCC in patients with elevated AFP before HCV treatment. Hence, based on our results and others [37], value of AFP > 10 detected during the pre-HCV treatment may indicate a closer surveillance of HCC. Nevertheless, more investigations are necessary to corroborate this finding.

In summary, the results of our investigation, which included a large prospective cohort of patients followed for five years after SVR in a rigorous clinical protocol, corroborate that the treatment with DAAs of cirrhotic HCV patients did not add risk to the development of HCC; in contrast, SVR was a protective factor. Nonetheless, cirrhotic patients with non-characterized nodules on ultrasound, high AFP levels in the pre-treatment period, and failure to achieve SVR were important risk factors for the development of HCC during or after HCV treatment with DAAs. In addition, although the results of our prospective study suggested that achieving SVR by DAA treatment reduced the incidence with time, the development of HCC still occurred, and the risk deserves careful attention of physicians who deal with these patients in their practice. As demonstrated here, the screening of HCC with abdominal ultrasound after cure of HCV infection favored the early diagnosis in more than 60% of patients who developed HCC.

An intrinsic limitation of this study is the absence of a control cohort of untreated HCV patients to accurately measure the reduction in the incidence of HCC. Nonetheless, our data showed that cirrhotic patients cured of HCV infection with DAAs had a yearly lower risk of development of HCC during follow-up.

## 5. Conclusions

In conclusion, our investigation adds scientific information to currently more accepted knowledge that the treatment with DAAs is associated with reduced risk for HCC. However, HCC still occurred, justifying the screening after cure of patients with advanced liver fibrosis, which favored early diagnosis in more than 60% of patients who developed the tumor. Some risk factors were predictable and can be identified to benefit patients with early diagnosis.

## Figures and Tables

**Figure 1 viruses-15-00221-f001:**
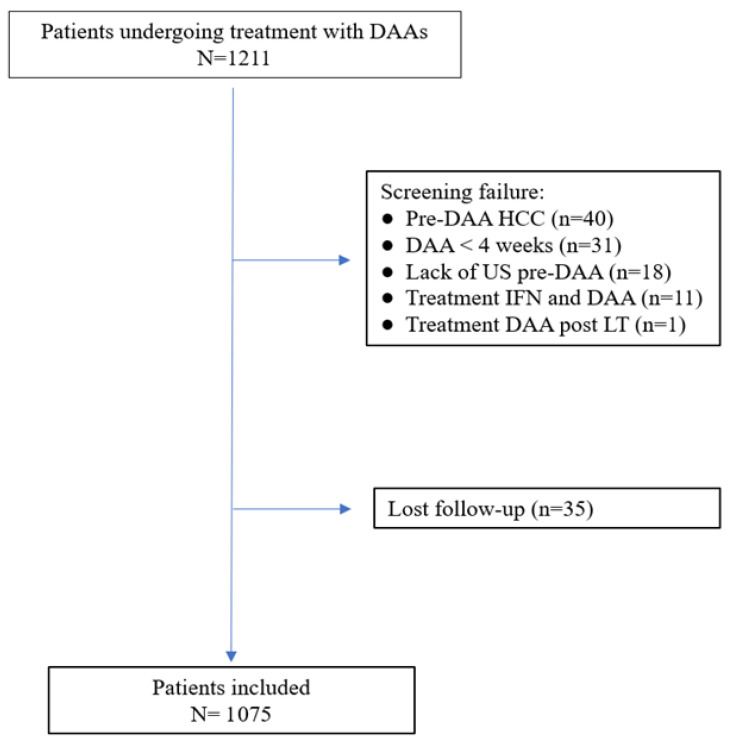
Flow chart of inclusion of patients in the study.

**Figure 2 viruses-15-00221-f002:**
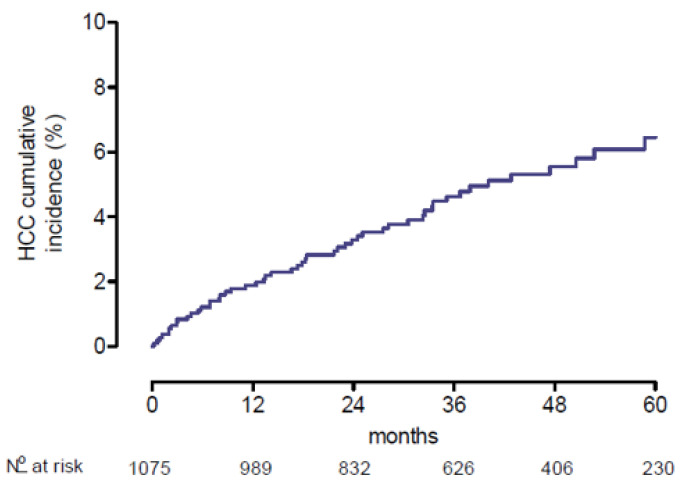
Five-year cumulative incidence of hepatocellular carcinoma (HCC) in patients with chronic hepatitis C treated with direct-acting antiviral drugs.

**Figure 3 viruses-15-00221-f003:**
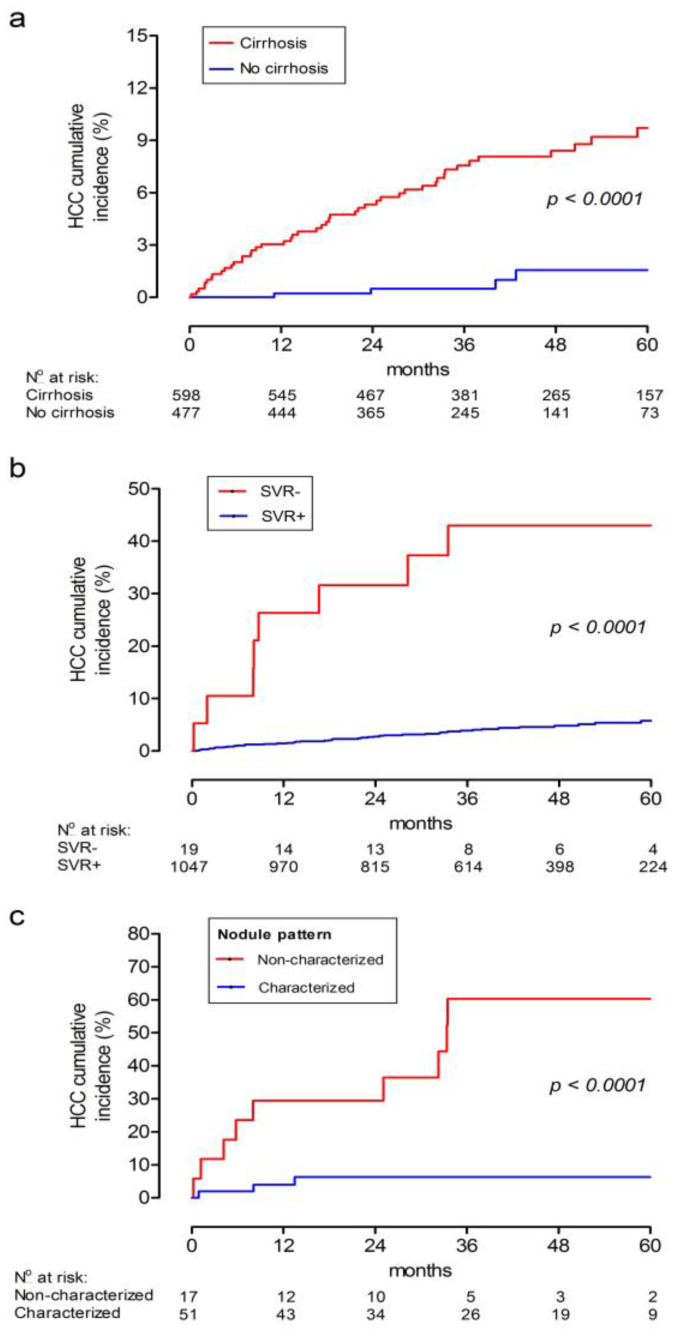
Cumulative incidence of HCC after the start of DAA treatment according to the presence of cirrhosis (**a**), non-SVR (**b**) and presence of non-characterized liver nodules (**c**). HCC: hepatocellular carcinoma; DAA: direct-acting antiviral agent; SVR: sustained virologic response.

**Figure 4 viruses-15-00221-f004:**
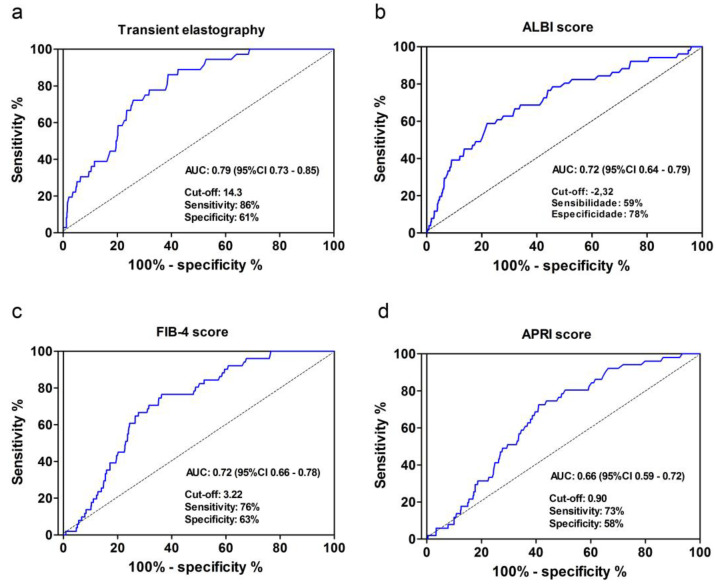
ROC curves of TE value (**a**), ALBI (**b**), FIB–4 (**c**) and APRI (**d**) before DAA treatment as predictors of the development of HCC. AUC: Area under the curve. ROC: receiver operating characteristic; TE: transient elastography; ALBI: albumin-bilirubin; FIB-4: fibrosis-4 index; APRI: aspartate aminotransferase to platelet ratio index; DAA: direct-acting antiviral agent; HCC: hepatocellular carcinoma.

**Figure 5 viruses-15-00221-f005:**
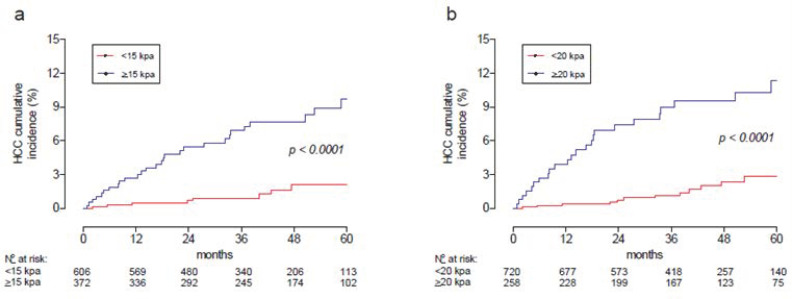
Incidence cumulative of HCC after the start of DAA treatment according to LSM <15 or ≥15 kPa (**a**) and <20 or ≥20 kPa (**b**) HCC: hepatocellular carcinoma; DAA: direct-acting antiviral agent.

**Table 1 viruses-15-00221-t001:** Baseline demographic, clinical and laboratorial characteristics of patients (N = 1075).

Female gender, *n* (%)	640 (59.5)
Age, years	60 (53–66)
Diabetes mellitus, *n* (%)	286 (26.6)
Alcohol use, *n* (%)	220 (20.5)
History of non-primary liver cancers, *n* (%)	57 (5.3)
Body mass index, kg/m^2^	26.1 (23.5–29.1)
ALBI score, *n* (%) *	
ALBI 1	615 (57.4)
ALBI 2	427 (39.8)
ALBI 3	30 (2.8)
Fibrosis stage, *n* (%)	
F0/F1	104 (9.7)
F2	171 (15.9)
F3	202 (18.8)
F4	598 (55.6)
Portal hypertension, *n* (%)	328 (30.5)
Ascites, *n* (%)	63 (5.9)
Pre-treatment liver nodules on ultrasound, *n* (%)	68 (6.3)
Characterized, *n* (%)	51 (4.7)
Non-characterized, *n* (%)	17 (1.6)
Coinfection,	93 (8.7)
HBV, *n* (%)	7 (0.7)
HIV, *n* (%)	86 (8.0)
LSM, kPa	12.1 (8.4–20.9)
APRI	0.78 (0.41–1.60)
FIB-4	2.53 (1.55–4.47)
AST, IU/L	52 (35–82)
ALT, IU/L	57 (38–95)
Total bilirubin (mg/dL)	0.72 (0.53–1.00)
GGT (IU/L)	82 (52–144)
Alkaline phosphatase (IU/L)	94 (70–115)
Serum albumin (g/dL)	4.0 (3.6–4.3)
Alpha-fetoprotein (ng/mL)	5.5 (3.3–10.2)
Platelet count (10^3^/mm^3^)	167 (115–224)
INR	1.08 (1.00–1.19)

* missing data (three patients). Values expressed as frequency (%) or median (interquartile range). ALBI: albumin–bilirubin; ALT: alanine aminotransferase; AST: aspartate aminotransferase; APRI: aspartate aminotransferase to platelet ratio index; DAA: direct-acting antivirals; FIB-4: fibrosis-4 index; GGT: gamma glutamyl transferase; HCV: hepatitis C virus; INR: international normalized ratio; LSM: liver stiffness measurement. Alcohol use was defined as alcohol consumption ≥ 30 g/day for women and ≥40 g/day for men.

**Table 2 viruses-15-00221-t002:** Univariate cox proportional hazard regression analysis of variables associated with the risk of developing hepatocellular carcinoma after the start of DAA treatment (N = 1075).

Variable	Univariate Analysis		
	HR Ratio	95% CI	*p*
Male gender	1.59	(0.91–2.78)	0.11
Age (years)	1.05	(1.02–1.08)	0.002
DM	1.18	(0.64–2.17)	0.60
BMI (kg/m^2^)	0.94	(0.88–1.01)	0.07
Alcohol use	2.52	(1.42–4.47)	0.002
LSM (kPa)	1.05	(1.03–1.07)	<0.0001
Non-characterized liver nodule	19.1	(9.25–39.44)	<0.0001
Ascites	7.88	(4.24–14.66)	<0.0001
Time since pretreatment imaging (months)	0.82	(0.63–1.07)	0.14
Serum albumin < 3.5 g/dL	3.76	(2.14–6.58)	<0.0001
Platelets < 100,000 mm^3^	3.00	(1.69–5.34)	<0.0001
Alpha-fetoprotein > 10 ng/mL	4.11	(2.32–7.26)	<0.0001
INR	1.99	(1.31–3.03)	0.001
Total bilirubin (mg/dL)	1.57	(1.22–2.03)	0.001
Alkaline phosphatase (IU/L)	1.01	(1.00–1.01)	0.013
GGT (IU/L)	1.00	(1.00–1.01)	0.25
Cirrhosis	11.50	(3.58–37.00)	<0.0001
Genotype 1	0.47	(0.24–0.92)	0.027
Non-SVR	12.50	(5.88–25.00)	<0.001

DAA: direct-acting antiviral agent; CI: confidence interval; DM: diabetes mellitus; BMI: body mass index; LSM: liver stiffness measurement; INR: international normalized ratio; GGT: gamma glutamyl transferase; SVR: sustained virologic response.

**Table 3 viruses-15-00221-t003:** Multivariate Cox proportional hazard regression analysis of variables associated with the risk of developing hepatocellular carcinoma after the start of DAA treatment (N = 1075).

Variable	Multivariate Analysis		
	Hazard Ratio	95% CI	*p*
Male gender	1.33	(0.57–3.10)	0.50
Age (years)	1.05	(1.01–1.09)	0.01
Alcohol use	1.49	(0.63–3.51)	0.37
Non-characterized nodule	10.92	(4.18–28.52)	<0.001
Time since pretreatment imaging (months)	1.11	(0.84–1.48)	0.46
LSM (kPa)	1.04	(1.02–1.06)	<0.001
Alpha-fetoprotein	1.001	(1.00–1.001)	0.007
Non-SVR	5.94	(1.90–18.54)	0.002

CI: confidence interval; LSM: liver stiffness measurement; SVR: sustained virologic response.

## Data Availability

Not applicable.

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
