# Peer review of "Incidence and Risk Factors of Hepatocellular Carcinoma in Patients with Chronic Hepatitis C Treated with Direct-Acting Antivirals"

_viruses, 2023, doi:10.3390/v15010221_

Round 1

Reviewer 1 Report

This prospective study investigated the HCC incidence and factors associated with HCC development following DAA treatment in CHC patients. The risk factor to HCC were non-characterized nodule, cirrhosis, AFP > 10 ng/mL and non-SVR. Authors concluded that HCV cure reduced particularly in cirrhotic patients. Overall, this manuscript is scientifically sound and of some novelty.

Comments:

1. Higher percentage of patients with non-characterized nodule before DAA treatment developed HCC within short-duration following HCV clearance. Authors also should explain why patients with non-characterized nodule developed HCC shorter than previous reports. 

2. Although authors described the features of non-characterized nodule in manuscript, under-diagnosis of HCC before enrolment is still an important concern. I would suggest authors to describe clearly how many patients received image studies or liver biopsy to exclude HCC in patients with non-characterized nodule before DAA treatment. 

3. In discussion, authors suggested that F3 patients did not require to receive HCC surveillance due to the limited medical resources. This point of view is controversial in different guideline recommendations. Discussion of comparisons among different guidelines is necessary. 

Reviewer 2 Report

DAA is a standard therapy for CHC, and the risk of HCC after DAA therapy is an important issue. The non-characterized liver nodule is a bright spot. However, some issues should be corrected.

[Major problem]

1. The distribution of ALBI grade is strange. According to the median of albumin and total bilirubin, the ALBI score = -2.68. According to the 3rd quartile of total bilirubin (1.0 mg/dL) and 1st quartile of albumin (3.6 g/dL), the ALBI grade = -2.25. There were 13.9% and 0.5% patients classified as Child-Pugh class B and C, respectively. Therefore, it seemed that the percentage of ABLI grade 2 or 3 should be more than 10%. 

2. It is wrong for the multivariable analysis including univariate variables with p <0.10. Please include univariate variables with p < 0.157 (Biom J. 2018;60:431) or < 0.200 (Am J Edpiemiol. 1993;138:923).

3. It is unreasonable to show the cumulative incidence of HCC according to the Child-Pugh class (Figure 3a). Because the Child-Pugh class is not a predictor of developing HCC in Table 2 or 3. 

4. Please describe the reason to select the cut-off of TE (14.3 kPa), ALBI (-3.95), FIB-4 (3.22), and APRI (0.90). 

5. If TE <15 or ≥ 15 kPa is a predictor of developing HCC (Figure 5), the variable (continuous TE or categorical TE) should be analyzed in the multivariable Cox regression analysis (Table 4). Please remember that K-M analysis is a univariate analysis. 

[Minior problem]

1. The patients who developed HCC before the end of DAA therapy should be excluded

2. Did TE have better predictability than ALBI, FIB-4, or APRI?

Reviewer 3 Report

Leal et al. presented a study incidence of HCC and associated risk factors in HCV patients treated with DAA. This is an interesting study enablig to identify key clinical factors related to HCC development after DAA treatment.

The study seems well conducted. However some questions remain.

1. The authors mention a follow-up of 5 years until February 2022 while the inclusion period is 2015-2019. Can they clarify this point and describe how the data from the patients enrolled in 2018-2019 have been included in the present study?

2. For the patients developping HCC after DAA treatment, without SVR, can the authors make a correlation between HCV viral load, HCV genoype, and HCC development quickness?

Round 2

Reviewer 2 Report

It is strange to pop up a new concept/value in the manuscript (eg. the best cut-off of TE = 14.3, but the cut-off = 15 or 20 in Figure 5). Continuous and reasonable logic is important.

[Major problems]

Points 1 to 2: Agree

Point 3: Figure 3a in the reply is different from figure 3a in the revised manuscript. Please modify Figure 3a in the revised manuscript or add Child B/C vs. Child A in Table 3.

Point 4: Please provide statistical evidence to support the cut-off. Younden's method could be considered.

Point 5: Please provide the evidence that collinearity between (continuous or categorical) TE and liver cirrhosis. An alternative method is to provide another multivariate Cox regression analysis using TE instead of cirrhosis (using categorical TE instead of cirrhosis in Table 3). By the way, it is better to use the same cut-off of TE in Figure 4a and Figure 5.
